# Developmental Dental Defects in Permanent Teeth Resulting from Trauma in Primary Dentition: A Systematic Review

**DOI:** 10.3390/ijerph19020754

**Published:** 2022-01-10

**Authors:** Lucía Caeiro-Villasenín, Clara Serna-Muñoz, Amparo Pérez-Silva, Ascensión Vicente-Hernández, Andrea Poza-Pascual, Antonio José Ortiz-Ruiz

**Affiliations:** 1Department of Integrated Pediatric Dentistry, School of Dentistry, University of Murcia, Biomedical Research Institute of Murcia, 30008 Murcia, Spain; luciacaeirovillasenin@gmail.com (L.C.-V.); perez_amparo@hotmail.com (A.P.-S.); ascenvi@um.es (A.V.-H.); ajortiz@um.es (A.J.O.-R.); 2Department of Stomatology I, School of Dentistry, University of the Basque Country, 48940 Lejona, Spain; poza.andrea@gmail.com

**Keywords:** dental trauma, permanent teeth, primary teeth, dental injury

## Abstract

The objective was to determine whether trauma in primary dentition causes alterations in the development of permanent dentition. Searches were made in May 2020 using PubMed, MEDLINE, MEDES, Scopus, Lilacs, and Embase. Papers in English, German, and Spanish, without restrictions in the year of publication, were included. The quality of the studies was analyzed using the NOS Scale. The search retrieved 537 references, and seven studies were included for a qualitative analysis. The results showed that trauma to a deciduous tooth can damage the bud of the permanent tooth. Enamel discoloration and/or hypoplasia were the most common sequelae in the permanent teeth after trauma to the primary predecessor. The type and severity of sequelae in the permanent tooth are associated with the development phase of the bud. Children with trauma of their primary teeth should receive checkups until the eruption of the permanent teeth for the early diagnosis and treatment of possible sequelae. Intrusion of the primary tooth was the trauma that caused the most damage and enamel alterations the most frequent sequelae.

## 1. Introduction

Children are especially vulnerable to dental trauma, especially in the first two years of life, when they are starting to walk and socialize. The prevalence of trauma ranges from 4% to 33% [1]. Epidemiological data show that approximately 30% of children aged <7 years have trauma in ≥1 primary incisor, and about 40% of children go to the dentist for the first time due to dental trauma [2].

Dental injuries have been recognized as an oral public health problem [3]. Trauma lesions are considered emergency situations as they require immediate attention and may have important medical, aesthetic, and psychological consequences for children and their parents [4].

Due to the close relationship between the apex of primary teeth and the bud of permanent teeth, any lesion to the primary dentition may influence the eruption of the permanent teeth [5].

The severity of sequelae depends on the patient’s age, the degree of root reabsorption, the type and extent of the trauma, and the degree of development of the permanent successor at the time of trauma. Intrusion and avulsion of primary teeth are considered the types of traumas that produce the greatest number of alterations in the development of permanent teeth, according to various studies [1,3,6,7,8].

The main consequences of primary tooth trauma in the development of the permanent teeth are enamel discoloration, enamel hypoplasia, coronal dilaceration, root dilaceration, odontoma-like malformations, and alterations in eruption [5,8,9]. Constant followup, with complementary tests, such as radiographs, and appropriate clinical interventions, can minimize or even prevent damage to the successor tooth [10,11].

The objective of this systematic review was to determine whether trauma in the primary dentition causes alterations in the development of permanent dentition.

The aims of the review were outlined using the components of the Patient, Intervention, Comparison, and Outcome (PICO) system [12]:

P: The patient population (or problem) addressed were permanent teeth whose predecessors suffered dental trauma.

I: The intervention or exposure of interest was the type of trauma to the primary teeth.

C: The comparators were permanent teeth whose primary predecessors had not had trauma.

O: The main result or end point of interest was complications in permanent teeth whose primary predecessor had had trauma.

## 2. Materials and Methods

The review was carried out in accordance with the Preferred Reporting Items for Systematic Reviews and Meta-analyses (PRISMA) statement on the publication of systematic reviews and meta-analyses [13] (Appendix A). The review was registered as CRD42019123188 at the Centre for Reviews and Dissemination, University of York, United Kingdom.

**Eligibility criteria.** In this review were included articles written in English, German, and Spanish languages whose authors linked trauma in human primary dentition to their consequences on the permanent dentition. We have excluded articles based in animal studies, pilot studies, editorials, letters, and literature reviews.

**Search and item selection strategy.** Without specifying the language, we performed a detailed search in the following electronic databases: PubMed, MEDLINE, MEDES, Scopus, Lilacs, and Embase. The search period was June 1972 [14] to May 2020 [15]. We used all terms related to dental trauma through the following search string, making modifications as necessary for the various database requirements: “primary dentition”, “permanent dentition”, “dental trauma”, “dental injury”, “root fracture”, “dentin-enamel fracture”, “enamel fracture”, “avulsion”, “intrusion”, “dislocation”, “subluxation”, and “concussion”.

Studies were selected in three phases (Figure 1). First, we considered only the title. Secondly, we considered the article abstract. If the abstract did not provide sufficient information to decide on study inclusion, we reviewed the full article before making a final decision. Articles written in languages other than English, Spanish, or German were discarded. Thirdly, we considered the full text of the article. The authors tried to obtain all available articles. Two authors (C.S., L.C.) carried out the three selection phases independently. Initially, there was a 93,7% of agreement, and the disagreements were solved by consensus. In cases where consensus was not reached, the authors consulted a third author (A.V.) who helped reach consensus.

**Data extraction and quality assessment.** The same two authors (C.S., L.C.) extracted data from the articles included and evaluated the quality of the studies using the Newcastle-Ottawa scale (NOS) [16] and cohort studies. For cross-sectional studies, the NOS tool for cohort studies was used.

According to the NOS scale, each study may be assigned a maximum score of 9, based on 3 different categories, classifying the study as “high quality” if the total score is ≥ 7. In our review, we were able to assign a maximum score of 8 since, in the “comparability” category, we considered a maximum score of one point, rather than two, because the studies analyzed related the exposed and unexposed groups by population only, without considering other factors. Therefore, we determined that these studies had to reach a score of 6 to be classified as “high quality”.

For some categories we had to determine appropriate cut-off points for evaluation. All authors agreed the following points: (a) a minimum of 50 children (first item in the selection category) was established as a representative sample, (b) the radiographic test (first item in the results category) was established as a common register for all studies and (c) an appropriate followup period up to the eruption of the permanent successor to the deciduous tooth with trauma (second item in the results category).

## 3. Results

The search strategy yielded 537 articles: 201 from Embase, 179 from Scopus, 92 from PubMed, 22 from Lilacs, and 43 from MEDLINE. No items were found in MEDES. After removing all duplicate items, 310 remained. After applying the selection criterion, we included 18 articles in the review (Figure 1) [1,3,4,5,6,7,8,10,11,17,18,19,20,21,22,23,24,25], including 13 cross-sectional studies [1,3,4,6,7,8,11,17,19,21,22,23,24], two cohort studies [20,25], and three case-control studies [5,10,18].

**Quality assessment.** Using the NOS scale for cohort and cross-sectional studies, we found that none of the 15 studies reached the maximum score of 8, but four with a score of ≥6 were classified as high quality [1,6,11,17] (Table 1). Using the NOS scale for case-control studies we found that, although none of the three studies reached the maximum score of 8, they could be classified as high quality [5,10,18] (Table 2). The remaining 10 studies were classified as low quality. Due to the lower evidence of the results in low quality articles, the systematic review only reports the results of the high-quality studies (Table 3 and Table 4).

**Basic results.** The studies were published over a period spanning almost 47 years. The first, by Andreasen et al. [5], was published in 1971 and the last, by Machado Lenzi et al. [10] in 2019. The studies came from various regions: three in Europe [1,5,18], two in Brazil [10,11], one in Turkey [6], and one in Switzerland [17].

Four were cross-sectional studies: observational and descriptive studies in which researchers called children who had had trauma in the primary dentition to evaluate the consequences in the permanent dentition [1,6,11,17]. The remaining three were case-control studies [5,10,18].

Table 3 and Table 4 summarize the data from high quality studies. The sample size varied from 78 to 879 children and 138 to 753 primary teeth with trauma. The children were aged between 0 and 17 years. All studies included children of both sexes: several found no differences between sexes in terms of the frequency of trauma [6,11] while others described a higher frequency in boys than in girls [1].

Most studies included all types of traumas [1,5,10,17], two only studied intrusions [6,11], and one specified the type of trauma [18]. The types of traumas were classified according to the Andreasen [26] in most studies reviewed. In most of the articles, the teeth most affected by trauma were the upper central incisors [1,6,10,11], consequences were described after intrusions [1,5,10,11,17].

In most studies, researchers reported the results as prevalence rates with a *p* value when there were significant differences. Andreasen et al. [5] between the type of trauma and the patient’s age at the time of trauma, with the consequences on permanent teeth being less frequent when trauma occurred in children aged > 4 years, results similar to those of Lenzi et al. [10]. Other studies found no significant relationship between the time of intrusion and sequelae in permanent dentition [6,11].

The prevalence of permanent tooth alterations ranged from 4.5% [1] to 68.8% [11], the most common being tooth enamel defects. Although many of the articles studied the consequences in both sets of teeth, we focused only on the consequences in permanent dentition.

### Consequences in Permanent Dentition in High Quality Articles

Hypoplasia and/or hypocalcification were the most common malformations in permanent teeth in all studies [1,5,6,10,11,18] especially after intrusion or avulsion. They included enamel discoloration, which ranged from white to yellow-brown, and defects of the enamel surface [17]. These lesions may be caused by environmental causes or genetic factors, emphasizing the need for a control group in studies [10].

Alterations in eruption were much less common than enamel lesions [1,11]. In the study by Altun et al. [6], ectopic eruption was observed as a single sequela in 23 teeth and combined with other sequelae in 7 teeth.

Coronal or root dilaceration was rare or infrequent [1,5,6,11]. Alterations in the development of the remaining teeth involving the crown occurred more frequently than those in which the root was involved [6,17].

Only one study mentioned two cases of hypomineralization due to injury to deciduous tooth [1]. Odontoma was rare or uncommon and was only observed in groups who had trauma, indicating a direct relationship [10].

## 4. Discussion

**Quality of studies.** The results indicate the need for further quality studies on the involvement of the permanent successor tooth following trauma in the primary dentition, since we were only able to include 7 of the 18 studies as high quality according to the NOS scale.

The main limitation of our study was the type of studies found in the systematic search. Due to the nature of the topic, the most common study design was cross-sectional. This is a type of observational study and, therefore, in a strict sense, the scientific evidence of the systematic review is not high. Since it is not possible to carry out randomized controlled clinical trials, perhaps prospective population-based cohort studies would be the most appropriate to learn more about the consequences of trauma in the primary dentition on permanent teeth.

We found differences in the study designs analyzed, the type of trauma analyzed, the age of the participants, the follow-up time, etc. Because of this heterogeneity between studies, we were unable to perform a meta-analysis. Instead, we validated the studies based on selection criteria, comparability, and the measurement of results according to the NOS scale (Table 1 and Table 2).

Of the seven high quality studies, only three had a control group [5,10,18]. There was a majority of observational studies, and the lack of a control group could have influenced the results, as the alterations in permanent dentition observed may be due to other causes (molar incisor hypomineralization, amelogenesis imperfecta, fluorosis, or dilaceration, which may be idiopathic), and not only because of the trauma in the primary dentition. A control group design would therefore be more appropriate and present fewer biases [10].

Machado Lenzi et al. [10] found that permanent teeth whose preceding primary teeth had trauma had a much higher risk of alterations when compared to the control group: 28.9% of the permanent teeth in the trauma group had alterations, while the prevalence of defects in the control group due to other causes was 7%. Andreasen et al. [18] found a high frequency of alterations in the group without previous trauma, suggesting there are nontraumatic factors involved in the etiology of these changes. However, the same authors also stated, in another study, that a nontrauma etiology probably does not explain more than 3% of the alterations [5].

**Consequences in permanent dentition after trauma in primary dentition.** The objective of this systematic review was to determine whether evidence in the literature that trauma in primary dentition causes alterations in the development of permanent succession teeth.

Any trauma to a primary tooth can damage the bud of the successor permanent tooth [17].

Discoloration of the enamel and/or hypoplasia were the most common sequelae in permanent teeth following trauma to its deciduous predecessor [1,4,5,6,7,8,10,11,17,21]. Several studies found that the predominance of enamel hypoplasia versus other developmental alterations is that it can be caused by less severe trauma in primary teeth [10,17].

Most mineralization defects are in the incisal half of the central and lateral incisors. In adjacent teeth, discoloration of the enamel may occur after being indirectly affected due to bleeding of the traumatized tooth [10,17].

The type and severity of sequelae in the permanent teeth were associated with the developmental phase of the bud. When the studies considered the development of the permanent tooth at the time of the injury, discoloration of the enamel appeared to occur in the early stages of the formation of both the crown and the root, while enamel discoloration associated with hypoplasia was only found in teeth injured during the formation of the crown [5,6]. Severe trauma to the permanent tooth bud at an early stage of odontogenesis may lead to complete deformation of the tooth, causing an odontoma-like formation [10].

Involvement of the crown occurred more often than root involvement or alterations in eruption. This may be attributed to the close relationship between the deciduous tooth root and the permanent tooth crown, and the fact that most traumatic injuries occur between one and four years of age, during the development of the permanent crown [6].

Some studies found that the types of traumas that cause the most sequelae are intrusions, followed by avulsions [6,20,22]. Von Arx et al. [17], found that more than half of cases with intrusive luxation developed permanent tooth malformations but found no alteration of the permanent tooth in any case of corono-radicular fracture. Andreasen et al. [5], stated that injury to the permanent tooth is evident, since the socket is fractured or compressed. In the case of avulsion, the slight rotational motion caused by the root curvature may injure the tissues that separate the primary tooth from the bud of the developing permanent tooth. Fracture of the alveolar bone, in addition to the dental injury, significantly increases the frequency of subsequent alterations in the permanent teeth.

Other studies, such as those by Guedes de Amorim et al. [8] and Ribeiro do Espírito Santo et al. [21], found no significant relationship between the type of trauma and the consequences in the permanent teeth.

Due to their position in the dental arch, the upper incisors are the teeth most affected by trauma. They are the most exposed teeth, especially in cases where they are protruding or there is lip incompetence [1,11,17]. The next most affected teeth are the upper and lower lateral incisors, and the upper canines, albeit with a large statistical difference [6].

Reports show that the severity of sequelae varies depending on the child’s age. Several studies analyzed the relationship between the child’s age at the time of trauma and sequelae in permanent teeth [4,5,24]. Damage secondary to trauma appears to be considerably greater when it occurs at a younger age. Studies report a higher percentage of permanent teeth abnormalities in patients aged < 2 years at the time of trauma [6,22]. A high risk of sequelae in this age group may be associated with incomplete bone and permanent teeth [4,8]. According to Von Arx et al. [17], except for enamel discoloration, all other types of developmental alterations were, to some extent, correlated with the time when the lesion occurred in the primary teeth. The fact that enamel mineralization maturation continues until the time of eruption explains why enamel discoloration may affect all age groups [7,10].

Machado Lenzi et al. [10] also found a lower prevalence of sequelae in children aged 5–7 years, while no 8-year-old with trauma presented sequelae.

Some studies found no correlation between the patient’s age at the time of trauma and the development of permanent tooth alterations [6,11].

Epidemiological studies of dental trauma provide important data on prevalence and associated factors, which may aid the development of clinical action and prevention protocols. Early treatment of trauma helps avoid further consequences on the tooth involved and its successor [11].

Although the seven included studies are of high quality according to the NOS assessment tool and include a total of 1067 children, with a minimum of 1911 primary teeth traumatized, the results of the review may be generalizable with caution to the involved population.

## 5. Conclusions

Children with dental trauma of the primary teeth should receive checkups for the diagnosis and treatment of possible sequelae until the eruption of the permanent teeth. The frequency of revisions will depend on the severity of the dental trauma, being more frequent the greater the severity.

Intrusion is the trauma that causes the most damage and alterations in enamel development the most frequent sequelae. The younger the patient, more serious can the damage to permanent tooth be.

More high-quality, prospective, controlled studies would be needed to reach a higher level of scientific evidence about the effects on permanent teeth of trauma in primary dentition.

## Figures and Tables

**Figure 1 ijerph-19-00754-f001:**
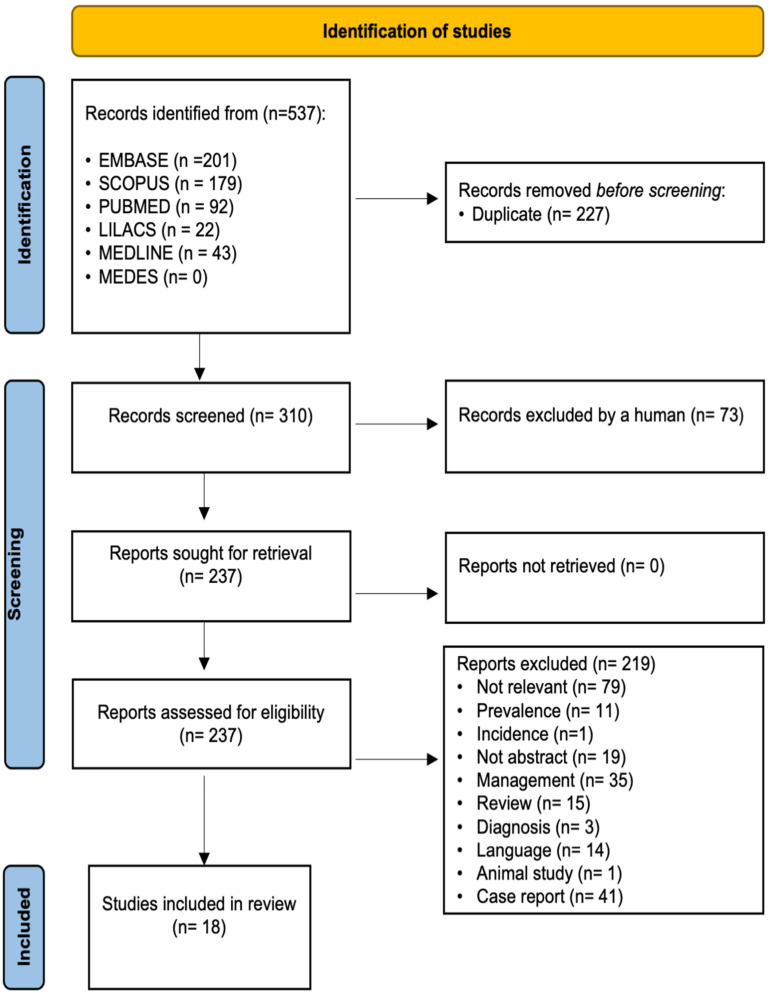
PRISMA Diagram.

**Table 1 ijerph-19-00754-t001:** Methodological quality for cross-sectional and cohort studies, assessed using the Newcastle-Ottawa Scale *.

Study	Country	Study Design	Criteria **								Total Score
			Selection				Comparability	Outcome and Exposure			
			1	2	3	4	5	6	7	8	
Von Arx and Colleagues, 1993 [17]	Switzerland	Cross-sectional	X	X	X	X		X	X	X	7
Odersjö and Colleagues, 2001 [19]	Sweden	Cross-sectional	X		X	X		X		X	5
Christophersen and Colleagues, 2005 [24]	Denmark	Cross- sectional	X		X	X			X		4
Sennhenn-Kirchner and Colleagues, 2006 [20]	Germany	Cohort	X		X	X		X			4
Altun and Colleagues, 2009 [6]	Turkey	Cross-sectional	X		X	X		X	X	X	6
Da Silva Assunção and Colleagues, 2009 [4]	Brazil	Cross- sectional	X		X	X		X	X		5
Ribeiro do Espírito Santo and Colleagues, 2009 [21]	Brazil	Cross-sectional	X		X	X		X	X		5
Carvalho and Colleagues, 2010 [11]	Brazil	Cross-sectional	X		X	X		X	X	X	6
Guedes de Amorim and Colleagues, 2010 [8]	Brazil	Cross-sectional	X		X	X		X	X		5
Cueto Urbina and Colleagues, 2012 [7]	Chile	Cross-sectional	X		X	X		X	X		5
Soares and Colleagues, 2014 [3]	Brazil	Cross-sectional	X		X	X					3
Mendoza-Mendoza and Colleagues, 2014 [1]	Spain	Cross-sectional	X		X	X		X	X	X	6
Bardellini and Colleagues, 2017 [22]	Italy	Cross-sectional	X		X	X		X	X		5
Silva de Amorim and Colleagues, 2018 [23]	Brazil	Cross-sectional	X		X	X		X	X		5
Graziele Martioli and Colleagues, 2019 [25]	Brazil	Cohort	X		X	X		X	X		5

* Source: Wells and Colleagues, 2000; ** We used the following criteria to assess the methodological quality of each study: representativeness of the exposed cohort (1); selection of the nonexposed cohort (2); ascertainment of exposure (3); demonstration that outcome of interest was not present at the start of study (4); comparability on the basis of confounding control in the design or analysis (5); assessment of outcome (6); duration of follow-up period (7); and adequacy of follow up (8). An “X” represents 1 point contributing to the total score, which represents the level of methodological quality we found for each study. Determining the methodological quality is important for determining the validity of the study results.

**Table 2 ijerph-19-00754-t002:** Methodological quality for case-control studies, assessed using the Newcastle-Ottawa Scale *.

Study	Country	Study Design	Criteria **								Total Score
			Selection				Comparability	Outcome and Exposure			
			1	2	3	4	5	6	7	8	
Andreasen and Colleagues, 1971 [5]	Denmark	Case-control	X		X	X		X	X	X	6
Andreasen and Colleagues, 1972 [18]	Denmark	Case-control	X	X	X		X	X	X	X	7
Machado Lenzi and Colleagues, 2018 [10]	Brazil	Case-control	X	X	X	X	X	X	X		7

***** Source: Wells and Colleagues, 2000; ****** We used the following criteria to assess the methodological quality of each study: adequate case definition (1); representativeness of the case participants (2); selection of control participants (3); definition of control participants (4); comparability on the basis of confounding control in the design or analysis (5); assessment of exposure (6); same methods for case control participants (7); and nonresponse rate (8). An “X” represents 1 point contributing to the total score. The total score represents the methodological quality we found for each study. Determining the methodological quality is important for determining the validity of the study results. Future researchers can note the placement of points for each study in this systematic review as a guideline for focusing the goals of future studies to increase the quality of the research on the topic.

**Table 3 ijerph-19-00754-t003:** Summary of the high-quality cohort and cross-sectional studies.

Study	Study Participants	Most Affected Teeth	Type of Primary Teeth Trauma	Most Damaging Trauma	Time When Trauma Occurred	Age at Dental Examination	Consequences in Permanent Dentition	Sequelae Prevalence	*p* Value
Von Arx and Colleagues, 1992 [17]	114 children (70 boys, 44 girls) 255 traumatized primary teeth Age 0–7 years	Central upper primary incisors (n = 161, 63%)	Intrusion (15%) Avulsion (18%) Partial luxation (40%) Subluxation (18%) Crow and root fractures with exposed pulp (3%) Crow and root fractures without exposed pulp (6%)	Intrusion (54% of cases developed malformations)	Mean age at the time of the trauma was 3.6 years old	Mean age at the time of re-examination 8.7 years	**Enamel hypoplasia** (68%) **Crown dilaceration** (17%) **Root malformation** (10%) **Odontoma-like teeth** (5%)	23% (n = 33)	*p* value NP
Altun and Colleagues, 2009 [6]	78 children (41 boys, 37 girls) 138 traumatized primary incisors Age 12–48 months	Maxillary incisors (93.47%, with right central primary incisors accounting for 41.3%)	Intrusion	Intrusion	Most injuries occurred between 13 and 36 months	Mean age 22.32 ± 9.72 months	**Enamel hypoplasia** (39 teeth, 28.3%) **Crown/root deformation** (23 teeth, 16.7%) **Ectopic eruption** (23 teeth, 16.7%)	53.6% (n = 74)	NS correlation between age of intrusion and frequency of subsequent developmental disturbances. (*p* > 0.05)
Carvalho and Colleagues, 2010 [11]	307 children (169 boys, 138 girls) 753 traumatized anterior deciduous teeth Age between 0–10 years old	Right central primary incisor (35.2% TI) (47.3% PI)	Intrusion (n = 221, 29.3%)	Intrusion	Children with ages from 1 to 4 years old were the most affected	NP	**Discoloration of enamel** TI (11.7%, n = 15) PI (12.9%, n = 12) **Enamel hypoplasia** TI (7.8%, n = 10) PI (15.1%, n = 14) **Crown dilaceration** TI (2.3%, n = 3) PI (3.2%, n = 3) **Root dilaceration** TI (1.6%, n = 2) PI (3.2%, n = 3) **Eruption disturbances** TI (6.3%, n = 8) PI (12.9%, n = 12) **Sequestration of permanent tooth germ** TI (0,8%, n = 1) PI (0%, n = 0)	68.8% (n = 84)	NS Correlation between the age of intrusion and the developmental disturbances on permanent teeth (*p* = 0.140)
Mendoza-Mendoza and Colleagues, 2014 [1]	879 children 191 had traumatic injury to the primary dentition (101 boys, 90 girls) Age 1–7 years old	Upper central primary incisors (86.9%)	Subluxation (47.29%) Intrusion (23.15%) Avulsion (13.63%) Lateral luxation (9.35%) Extrusive luxation (5.9%) Hard tissue lesion (31.64%)	Intrusion	Most common age range for injuries in deciduous teeth was 1–3 years old	43 children: 1 year old 57 children: 2 years old 42 children: 3 years old 22 children: 4 years old 27 children: 5 years old or more.	**Hypoplasia** (2 cases) **Hypomineralization** (2 cases) **Delayed eruption** (2 Case) **Others** (2 cases)	4.5 %	*p* value NP

NP: Not provided; NS: not significant; TI: Total intrusion; PI: Partial intrusion.

**Table 4 ijerph-19-00754-t004:** Summary of the high-quality case-control studies.

Study	Study Participants	Most Affected Teeth	Kind of Primary Teeth Trauma	Most Damaging Trauma	Time when Trauma Occurred	Age at Dental Examination	Consequences in Permanent Dentition	Sequelae Prevalence	*p* Value
Andreasen and Colleagues, 1971 [5]	**TG:** 103 patients 213 traumatized primary teeth **CG:** Contralateral permanent successors 26 children 33 teeth Age 0–9 years old	Maxillary central primary incisors (n = 131)	Subluxation: 35 primary teeth Intrusive luxation: 36 primary teeth Extrusive luxation: 76 primary teeth Exarticulation: 27 primary teeth No information: 39 primary teeth	Intrusion (69%, n = 25)	62 children: 0 to 2 years old 43 children: 3 to 4 years old 88 children: 5 to 6 years old 20 children: 7 to 9 years old	NP	**White or yellow- brown discoloration of enamel** (23%) **White or yellow-brown discoloration of enamel and circular enamel hypoplasia** (12%) **Crown dilaceration** (3%) **Lateral root angulation or dilaceration** (1%) **Partial or complete arrest of root formation** (2%) **Not disturbances** (59%)	TG: 41% (n = 88)	*p* ≤ 0.05
Andreasen and Colleagues, 1973 [18]	**Main material:** 487 children (251 boys, 236 girls) **TG:** 147 children **NTG:** 340 children **CG:** 111 children (51 boys, 60 girls) Age 9–17 years old	Only anterior teeth were included	Luxations and fractures	NP	NP	NP	**Internal white enamel hypoplasia < 0.5mm** TG: 19.7% NTG: 21.2% **Internal white enamel hypoplasia ≥ 0.5mm** TG: 19.0% NTG: 13.5% **Internal white and yellow-brown enamel hypoplasia** TG: 2.7% NTG:1.8% **External white enamel hypoplasia** TG: 3.4% NTG:2.3% **External white and yellow-brown enamel hypoplasia** TG: 5.4% NTG:0.6% **White and/or yellow-brown discoloration of enamel and horizontal enamel hypoplasia** TG: 2.0% NTG:0.6% **Generalized internal or external white enamel hypoplasia** TG: 5.4% NTG: 5.3%	Main material: TG: 57.8% NTG:45.3%	*p* ≤ 0.05
Machado Lenzi and Colleagues, 2018 [10]	124 children **TG:** (permanent teeth whose antecessor had suffered dental trauma): 214 primary teeth **CG:** (teeth of the same child whose antecessor had not suffered dental trauma): 247 primary teeth Age 0–8 years old	Upper central primary incisors (n = 172, 80%)	Intrusion (38.7%) Concussion (14%) Lateral luxation (11%) Avulsion (11%) Subluxation (7.8%) Extrusion (4.6%) Enamel fracture (3%) Root fracture (3%) Enamel dentin fracture with pulp exposure (1.5%)	Intrusion (55.8% of cases developed malformations)	Enamel fracture mainly for the 2–3 years age group Enamel dentin fracture without pulp exposure mostly for the 1–2 years age group Enamel dentin fracture with pulp exposure mainly for the 3–5 years age group Root fracture mostly for the 4–5 years age group	<1 year: 1 child 1 year: 8 children 2 year: 49 children 3 year: 35 children 4 year: 22 children 5 year: 54 children 6 year: 28 children 7 year: 14 children 8 year: 3 children	**Discoloration of enamel** CG (5.7%) TG (11.2%) **Enamel hypoplasia** CG (1.2%) TG (9.8%) **Crown dilaceration** CG (0%) TG (0.9%) **Odontoma-like formation** CG (0%) TG (0.5%) **Root dilacerations** CG (0.4%) TG (0.9 %) **Partial arrest of root formations** CG (0%) TG (0.5%) **Sequestration of tooth germ** CG (0%) TG (0.5%) **Eruption disturbance** CG (0%) TG (4.7%)	TG: 28.9% (n = 62)	<0.001 *for intrusion* <0.001 *age* < *1 year*

NP: Not provided; TG: Trauma group; CG: Control group; NTG: Non-trauma group.

## Data Availability

The datasets used for the current study are available from the corresponding author upon reasonable request.

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
