# Peer review of "Developmental Dental Defects in Permanent Teeth Resulting from Trauma in Primary Dentition: A Systematic Review"

_ijerph, 2022, doi:10.3390/ijerph19020754_

Round 1
Reviewer 1 Report
Please open the attached file

Reviewer 2 Report
This article dealt with developmental defect in permanent teeth resulting from trauma to primary teeth.
I have a few questions and suggestions.
- Primary teeth to be a more appropriate term than temporary teeth.
- line 43 Intrusion and avulsion of temporary teeth are considered the types of trauma that have the most consequences in permanent teeth
What does ‘most consequences’ mean?
- line 188, Of the seven high quality studies only three had a control group.
Then, how did four studies without controls compare with non-traumatic teeth?
Reviewer 3 Report
The manuscript submitted by Lucía Caeiro Villasenín et al, reports the effects of the temporary tooth trauma on the permanent dentition.
The manuscript itself is well constructed and clear.
The topic is very interesting and adequate for this journal.
The English is clear and spelled correctly.
Regarding the title, in my opinion it should be changed. The primary tooth trauma should be changed, because when somebody reads the title he/she thinks of the occlusal trauma.
The introduction is well organised and clear, but you could provide more information regarding the relationship between the temporary tooth trauma and the permanent dentition.
I do not have any comments regarding the methodology and significance of this important issue covered in this review.
The discussion is well-conducted, supported by the articles for the literature.
The conclusion are correctly.
The author should explain clearly the primary tooth trauma or the word should be replaced (in my opinion primary trauma means occlusal trauma).
Reference format should be standardized.
It would be useful for readers of this paper to know the author’s personal experience regarding the consequence of temporary tooth trauma.
